# Cry4Aa and Cry4Ba Mosquito-Active Toxins Utilize Different Domains in Binding to a Particular *Culex* ALP Isoform: A Functional Toxin Receptor Implicating Differential Actions on Target Larvae

**DOI:** 10.3390/toxins14100652

**Published:** 2022-09-21

**Authors:** Manussawee Dechkla, Sathapat Charoenjotivadhanakul, Chompounoot Imtong, Sarinporn Visitsattapongse, Hui-Chun Li, Chanan Angsuthanasombat

**Affiliations:** 1Department of Environmental Biology, Faculty of Science and Technology, Suan Sunandha Rajabhat University, Bangkok 10300, Thailand; 2Bacterial Toxin Research Innovation Cluster (BRIC), Institute of Molecular Biosciences, Salaya Campus, Mahidol University, Nakorn Pathom 73170, Thailand; 3Laboratory of Structural Biochemistry and Cell Chemical Biology, Biophysics Institute for Research and Development (BIRD), Fang, Chiang Mai 50110, Thailand; 4Department of Biomedical Engineering, School of Engineering, King Mongkut’s Institute of Technology Ladkrabang, Ladkrabang, Bangkok 10520, Thailand; 5Department of Biochemistry, School of Medicine, Tzu Chi University, Hualien 97004, Taiwan

**Keywords:** Cry4 mosquito-active toxins, cytotoxicity assay, immunolocalization, GPI-anchorALPs, molecular docking, binding affinity change

## Abstract

The three-domain Cry4Aa toxin produced from *Bacillus thuringiensis* subsp. *israelensis* was previously shown to be much more toxic to *Culex* mosquito larvae than its closely related toxin—Cry4Ba. The interaction of these two individual toxins with target receptors on susceptible larval midgut cells is likely to be the critical determinant in their differential toxicity. Here, two full-length membrane-bound alkaline phosphatase (mALP) isoforms from *Culex quinquefasciatus* larvae, *Cq-*mALP1263and *Cq-*mALP1264, predicted to be GPI-linked was cloned and functionally expressed in *Spodoptera frugiperda* (*Sf*9) cells as 57- and 61-kDa membrane-bound proteins, respectively. Bioinformatics analysis disclosed that both *Cq-*mALP isoforms share significant sequence similarity to *Aedes aegypti*-mALP—a Cry4Ba toxin receptor. In cytotoxicity assays, *Sf*9 cells expressing *Cq-*mALP1264, but not *Cq-*mALP1263, showed remarkably greater susceptibility to Cry4Aa than Cry4Ba, while immunolocalization studies revealed that both toxins were capable of binding to each *Cq-*mALP expressed on the cell membrane surface. Molecular docking of the *Cq*-mALP1264-modeled structure with individual Cry4 toxins revealed that Cry4Aa could bind to *Cq*-mALP1264 primarily through particular residues on three surface-exposed loops in the receptor-binding domain—DII, including Thr^512^, Tyr^513^ and Lys^514^ in the β10-β11loop. Dissimilarly, Cry4Ba appeared to utilize only certain residues in its C-terminal domain—DIII to interact with such a *Culex* counterpart receptor. Ala-substitutions of selected β10-β11loop residues (T512A, Y513A and K514A) revealed that only the K514A mutant displayed a drastic decrease in biotoxicity against *C. quinquefasciatus* larvae. Further substitution of Lys^514^ with Asp (K514D) revealed a further decrease in larval toxicity. Furthermore, in silico calculation of the binding affinity change (ΔΔG_bind_) in Cry4Aa-*Cq*-mALP1264 interactions upon these single-substitutions revealed that the K514D mutation displayed the largest ΔΔG_bind_ value as compared to three other mutations, signifying an adverse impact of a negative charge at this critical receptor-binding position. Altogether, our present study has disclosed that these two related-Cry4 mosquito-active toxins conceivably exploited different domains in functional binding to the same *Culex* membrane-bound ALP isoform—*Cq-*mALP1264 for mediating differential toxicity against *Culex* target larvae.

## 1. Introduction 

Over the past few decades, *Bacillus thuringiensis* subsp. *israelensis*(*Bti*)—a Gram-positive, spore-forming bacterium, has become one of the most promising bio-insecticides for the control of mosquitoes-human disease vectors, which annually transmit parasitic and viral infectious diseases in millions of lives worldwide [1,2,3]. During sporulation, *Bti*produces insecticidal crystal proteins in the form of inclusions comprising both crystalline (Cry) and cytolytic (Cyt) δ-endotoxins, including Cry4Aa (~130 kDa), Cry4Ba (~130 kDa), Cry11Aa (~65 kDa) and Cyt1Aa (~27 kDa), which are, to various extents, toxic to mosquito larvae of three medically important genera, i.e., *Aedes*, *Anopheles* and *Culex* [2,3]. For example, the two closely related 130-kDa *Bti*toxins–Cry4Aa and Cry4Ba exhibit comparable toxicity against both *Aedes* and *Anopheles* larvae, but the Cry4Aa toxin is much more active against *Culex* larvae [3,4].

Individual native Cry toxins are primarily synthesized as inactive protoxins found within the parasporal crystalline inclusions. After being ingested by susceptible insect larvae, the protoxin inclusions are solubilized in alkaline midgut lumen and subsequently processed by gut proteases to yield active toxins of ~65 kDa [5,6]. The activated toxins are believed to first bind to specific membrane components (so-called receptors) lining the brush-border membrane of larval midgut epithelial cells [7], followed by toxin insertion into the target cell membrane to form an ion-leakage pore, which eventually results in midgut cell lysis. Disruption of the gut epithelium would lead to starvation and finally to the death of intoxicated larvae [8]. However, structural insights into toxin-receptor interactions underlying specific toxic effects of individual Cry toxins remain to be clearly elucidated.

Currently, the X-ray crystal structures of numerous Cry toxins [9,10,11,12,13,14], including the two mosquito-active toxins—Cry4Aa and Cry4Ba [13,14] have been solved. All these 65-kDa activated toxins display a typical wedge-shaped arrangement of three structurally different domains: an N-terminal domain of anti-parallel α-helical bundle(DI), a middle domain of three-β-sheet prism (DII) and a C-terminal domain of β-sheet sandwich (DIII) [13,14] (see Figure 1). Despite the fact that both Cry4Aa and Cry4Ba structures bear a resemblance to the other known Crystructures, their finer features are rather different since there is additional in vitro proteolysis by trypsin occurring at Cry4Aa-Arg^235^ as well as Cry4Ba-Arg^203^ located in the α5-α6 loop within DI [15]. Such tryptic cleavages would produce two trypsin-resistant fragments of ~47 and ~20 kDa that remain non-covalently associated to form the 65-kDa active toxin [15]. Accordingly, the trypsin-cleavage site at either Cry4Aa-Arg^235^ or Cry4Ba-Arg^203^ was eliminated *via* Gln-substitution to generate a 65-kDa trypsin-activated toxin fragment, thus being beneficial for protein crystallography as reported previously [16,17].

Thus far, both DI(the N-terminal α-helical bundle) and DII (the β-sheet prism structure with several exposed loops) of various Cry toxins have been evidently shown to play an important role in pore formation and specific target recognition, respectively [8,18,19,20]. Of particular significant findings in DI, we have given direct evidence for membrane-perturbing activity of the pore-lining α4-loop-α5 hairpin purified from the Cry4Ba toxin [21]. Recently, we have also denoted a crucial involvement in Cry4Aa biotoxicity of His^180^ located in the pore-lumen-facing α4 [22]. For a functional role in receptor recognition of DII, although most other studies are restricted to only three β-hairpin loops, i.e.,β2-β3, β6-β7 and β10-β11loops (originally assigned as loops 1, 2 and 3, respectively [9]), we have demonstrated that two other Cry4Ba-loops, i.e., β4-β5 and β8-β9loops, also play a critical role in receptor binding, and hence larval toxicity [23,24]. Moreover, we have shown that charge-reversal mutations at Asp^454^located within the β10-β11loop ofCry4Ba-DII noticeably enhance the toxin activity against less susceptible *Culex* larvae, suggesting a role of the charged side-chain in determining target specificity [25]. We have also disclosed that the structural stability of two receptor-binding hairpins (*i.e*. β2-β3 and β4-β5 within Cry4Ba-DII) through H-bonding between Thr^328^-Thr^369^ side-chains plays an important role in toxin binding to the *Bt*-Cyt2Aa2 toxin—an alternative receptor for Cry4Ba [26]. While various studies have suggested that the C-terminal domain—DIII could be implicated in preserving structural integrity or in determining target insect specificity [27,28,29,30,31,32,33], we have recently revealed that Cry4Ba-DIII could function as a tight-binding anchor for lipid membrane bilayers, highlighting its potential contribution to toxin-membrane interactions to mediate toxin activity [34]. More recently, we have demonstrated that Cry4Ba-DIII could also act as a receptor-binding moiety, signifying another contribution to toxin interactions with its target protein receptor—*Aa*-mALP in mediating larval toxicity [35].

Of particular interest, the specific interaction of Cry toxins with their different types of larval midgut receptors is recognized as an essential step leading to their lethal action. Up to now, numerous different membrane proteins from various target larval midguts have been identified and verified as potential Cry toxin receptors, including cadherin-like proteins, glycosyl phosphatidylinositol (GPI)-anchored aminopeptidases-N (APNs), GPI-anchored ALPs and ATP- binding cassette subfamily C member 2 transporter proteins (ABCC2) [7,36,37,38,39,40,41,42,43]. Previously, we have succeeded in identifying two different types of Cry4Ba-specifc receptors from *A. aegypti* mosquito larvae, i.e., GPI-anchored ALP and GPI-anchored APN [44,45].In the present study, two GPI-anchored ALP isoforms from *C. quinquefasciatus* (designated as *Cq-*mALP1263and *Cq-*mALP1264) were cloned, heterologously expressed in *S. frugiperda* (*Sf*9) cells and functionally characterized for the interaction with two closely-related Cry4 toxins—Cry4Aa and Cry4Ba. Differential susceptibility to individual toxins of *Sf*9 cells expressing each *Cq*-mALP was evidently revealed by cytotoxicity and immunolocalization assays. In silico analyses *via* homology-based modeling and molecular docking conceivably revealed the distinct characteristics of both toxins in binding to one particular functional isoform—*Cq-*mALP1264, supporting the fact that these two closely related toxins are toxic to *Culex* mosquito-larvae at greatly different degrees of toxicity.

## 2. Results 

### 2.1. Typical Features of Membrane-Bound ALP Isoforms from Culex Larval Midguts

Two different full-length genes encoding membrane-bound ALP homologues designated as *Cq*-mALP1263and *Cq*-mALP1264 were cloned from *C. quinquefasciatus* larval midgut RNA transcripts with specific primers using the RT-PCR strategy. The cDNA products obtained were of the expected sizes, i.e.,1569 bp for *Cq*-mALP1263 and 1656 bp for *Cq*-mALP1264 (Figure 2, Inset). The translated sequences of *Cq*-mALP1263 and *Cq*-mALP1264 revealed open reading frames of 523 and 552 amino acid residues (Appendix A) with a predicted molecular mass of ~57 and ~61 kDa, respectively. The ALP conserved sequences, including residues for substrate- and metal-binding sites in the catalytic domain [46,47], were identified in both *Cq*-mALP clones (Figure 2, *lower panel*), indicating that they would belong to the ALP family. In addition, both *Cq*-mALPs also contained a putative N-terminal signal peptide together with their predicted GPI-anchored sites on the C-terminus (Figure 2, *upperpanel**)*, suggesting that these two *Cq*-mALP homologues were produced as membrane-bound proteins attached to the cell surface *via* a GPI-anchor.

### 2.2. High Levels of Surface Expression of Cq-mALPs in Sf9 Insect Cells and Their Phosphatase Activity 

Upon baculovirus expression under the control of the strong polyhedrin promotor—AcMNPV (*Autographa californica* multiple nuclear polyhedrosis virus), the recombinant *Cq*-mALP proteins were highly produced in the *Sf9* infected cells after a 3-day post-infection as analyzed by SDS-PAGE (sodium dodecyl sulphate-polyacrylamide gel electrophoresis) (Figure 3A). Both *Cq-*mALP1263 and *Cq-*mALP1264 were identified as dominant protein bands with sizes corresponding to the predicted mass of ~57 and ~61 kDa, respectively. For the control cells, a ~30-kDa protein band was observed as expected for CAT (chloramphenicol acetyltransferase). 

Further verification *via* LC-MS/MS confirmed that the 57- and 61-kDa protein bands definitely represent the expressed products of *Cq-*mALP1263 and *Cq-*mALP1264, respectively, as multiple trypsin-generated peptide sequences were found exactly match with the deduced amino acid sequences of both *Cq*-mALP1263 (^44^STLAQQEQTIEYWRDNAK^61^, ^80^NIILFLGDGMSISTVAMAR^98^, ^226^DVAQQLVHGETGKRLKVVMGGGRREFLSSTLDPETGKK^263^ and ^460^MDMTADDFR^468^) and *Cq*-mALP1264 (^41^QVVGYEEIETTSDFWRQKAQSILQGK^66^, ^78^NVIYFIGDG MSPQTVAATRVYLGNE NRMLSFEEFPYIGTAR^118^, ^163^EETEFLGLLKWAQNEMATGVVSNARITHATPAGTYASIANR^204^ and ^343^RETGYVLFVEGGKIDMAHHETHPRLALE ETAEYSR^377^).

Additionally, ALP activity of individual *Cq-*mALP-expressed cell lysate was determined spectrophotometrically for the ability to hydrolyze pNPP (p-nitrophenyl phosphate—a phosphatase substrate) top-nitrophenol. The results shown in Figure 3B revealed that both *Cq-*mALP-expressed cells produced significantly higher ALP activity than CAT-expressed cells. Of course, endogenous ALP activity was also detected in non-*Cq-*mALP-expressed cells; it should be noted that the phosphatase activity of the *Cq-*mALP1264 lysate appeared to be lower than that of the *Cq-*mALP1263 lysate.

### 2.3. Cytotoxic Effects of Cry4 Toxins against Sf9 Cells Mediated by Individual Cq-mALP Isoforms

In vitro cytotoxicity assays were carried out to test whether the two recombinant *Cq*-mALP homologues can serve as a functional receptor for each Cry4 toxin.The change in morphological features of Cry4 toxin-treated *Sf*9 cells was examined under *an inverted* phase-contrast *microscope* in comparison with the control groups (see Figure 4A).For Cry4Aa-treated cells, only the *Cq*-mALP1264-expressed cells showed the most significant change in cell morphology as undergoing cell swelling and lysis when compared to unaffected effects on *Cq*-mALP1263- and CAT-expressed cells (Figure 4A, *upper panels*). Similarly, when treated with Cry4Ba, *Cq*-mALP1264-expressed cells displayed a significant morphological change while both the *Cq*-mALP1263-expressed cells and the control groups were completely unaffected (Figure 4A, *lower panels*).

Attempts were also made to quantitate this observation *via* cell viability assays. For the Cry4Aa-treated cells, the result revealed an increase in percent mortality of the *Sf*9cells expressing *Cq*-mALP1264 (~50% mortality) when compared to the treated cells expressing either *Cq*-mALP1263 or CAT that both showed cell mortality of less than 10% (Figure 4B). However, under the Cry4Ba treatment, a lesser extent (~20% mortality) was observed for the *Cq*-mALP1264-expressed cells while both *Cq*-mALP1263- and CAT-expressed cells appeared to be unaffected (see Figure 4B). Taken together, individual cytotoxic results clearly revealed that both Cry4Aa and Cry4Ba toxins could exploit only *Cq*-mALP1264, one of the two cloned *Cq*-mALP isoforms, as a functional receptor in mediating cell lysis. In addition, *Cq*-mALP1264-expressed cells appeared to be much more susceptible to Cry4Aa than Cry4Ba. 

### 2.4. Cell-Surface Localization of Cry4 Toxins on Cq-mALP-Expressed Cells

Further studies *via* immunofluorescence experiments revealed that Cry4Aa-localization through immuno-reactivity was clearly detected on the cell membrane surface of both *Cq*-mALP1263- and *Cq*-mALP1264-expressed *Sf*9cells, but immuno-staining patterns of *Cq*-mALP1264-expressed cells were observed with greater visible fluorescence (Figure 5A). For the Cry4Ba-treated cells, similar immuno-staining patterns of *Cq*-mALP12643- and *Cq*-mALP1264- expressed cells were observed, albeit the level of Cry4Ba staining appeared to be much lower in the *Cq*-mALP1263-expressed cells as expected (Figure 5B). Conversely, no detectable immuno-reactivity was observed in the control CAT-expressed cells, indicating that both Cry4Aa and Cry4Ba were able to bind only to *Sf*9 cells-expressing membrane-bound *Cq*-mALPs; these results also indicated that both *Cq*-mALP isoforms were predominantly expressed and localized on the membrane surface of *Sf9* cells.

### 2.5. Structural Features of Cq-mALP Isoforms and Implications for Specific Interactions

Further attempts were made to construct a plausible homology-based model of both *Cq-*mALP homologues, which were built based on the highest sequence similarity of the known shrimp-ALP structure (PDB: 1K7H) (see Appendix A). The resulting Ramachandran plot with the Phi/Psi values observed for the modeled structure revealed that >98% of the non-Gly and non-Pro residues have the backbone-dihedral angles in the energetically favorable and allowed regions, indicating that both 3D-modeled structures would remain in energetically and sterically favorable main-chain conformations. In addition, the Ramachandran *z*-scores, which characterize the shape of the torsion angle distribution in the plot of both modeled structures (i.e., −1.95 for *Cq-*mALP1263 and −1.97 for *Cq-*mALP1264) is in the range of scores for all determined structures at a similar size and quite similar to that of the shrimp-ALP template. Moreover, C_α_-trace superposition of both *Cq*-mALP models with the shrimp-ALP template revealed that *Cq*-mALP1263 shows a 0.37-Å RMSD for 456 equivalent C_α_ atoms out of 473 C_α_ atoms while *Cq*-mALP1264 displays a 0.25-Å RMSD for 450 equivalent C_α_ atoms out of 474 C_α_ atoms; these results would indicate a very high structural similarity in their 3D folds (Figure 6A).

As illustrated in Figure 6B, the central portion of both *Cq*-mALP modeled structures comprises a β-sheet core of ten β-strands, all but one, i.e., β15, are parallel, connected by α-helices to form a two-layer α/β sandwich, which is a typical topology of α/β hydrolase family [48]; it can also be inferred from Figure 6C that several uncharged-polar and charged residues, e.g., Lys^58^, Ser^61^, Thr^74^, Arg^162^, Glu^166^, Asp^208^, Tyr^211^, Asp^380^ Lys^410^, Glu^438^, Asp^506^, Glu^513^ and Tyr^517^, which are preserved exclusively in the functional *Cq-*mALP1264 molecule (see Appendix A), are found particularly on its surface-exposed area; these residues may perhaps participate in interactions of this receptor with its counterpart ligands—Cry4Aa and Cry4Ba although these non-conserved residues might be responsible for upholding other features.

### 2.6. Cry4 Toxin-Cq-mALP Docking Complex with Potential Interacting Residues

In silico molecular docking was subsequently performed to gain more critical insights into a particular architectural complex of individual Cry4 toxins interacting with their functional receptor—*Cq*-mALP1264 or non-functional receptor—*Cq*-mALP1263. As illustrated in Figure 7, the docking results revealed that Cry4Aa bound to *Cq*-mALP1264 or*Cq*-mALP1263 primarily through the three surface-exposed loops of the receptor-binding domain—DII, while Cry4Ba exploited merely its C-terminal domain—DIII to interact with both *Cq*-mALP isoforms.

As can be seen that the Cry4Aa-DII binding to *Cq*-mALP1264 is mainly mediated through residues in β2-β3 (Asn^377^), β6-β7 (Lys^432^, Tyr^433^ and Asp^436^) and β10-β11 (Thr^512^, Tyr^513^ and Lys^514^) loops (see Figure 1A and Figure 7A). Dissimilarly, it turned out that Cry4Ba-DIII, not DII, could interact with *Cq*-mALP1264 only through a patch of seven residues, i.e., Arg^520^ andGlu^522^ in β16, Lys^526^ and Asp^576^ in the β16-β17 loop, Ser^603^ and Gln^606^ in the β21-β22loop and Gln^635^ inα9 (see Figure 1B and Figure 7B). Altogether, such docking results of Cry4-*Cq*-mALP complexes indicated that the different domain regions, i.e., Cry4Aa-DII and Cry4Ba-DIII, would be individually responsible for functional binding to the same single receptor—*Cq*-mALP1264. In the toxin-*Cq*-mALP1263 docking complexes, Cry4Aa also used the same three surface-exposed loops in DII through residues in β2-β3 (Thr^375^), β6-β7 (Asn^431^) and β10-β11 (Thr^512^) loops in binding to the non-functional isoform (see Figure 7C) while Cry4Ba utilized a patch of five residues in DIII, i.e., Glu^522^ in β16, Lys^526^ and Arg^562^ in the β16-β17 loop, and Arg^601^ and Asn^605^ in the β21-β22loop in such an interaction (see Figure 7D). 

### 2.7. Biotoxicity Impairment of Cry4Aa Caused by Ala-Substitution of DII-Lys^514^

It was previously shown that the replacement of a few residues in the β10-β11 loop of Cry4Ba with corresponding loop residues of Cry4Aa caused significant increases in toxicity of greater than ~700-fold against *C. quinquefasciatus* larvae [20]. Herein, the three predicted binding residues on the β10-β11 loop in Cry4Aa-DII (i.e., Thr^512^, Tyr^513^ and Lys^514^) were thus selected accordingly for testing their possible involvement in larval toxicity. All three Ala-substituted mutant toxins, T512A, Y513A and Y514A, were successfully generated as seeing that each mutant was over-expressed in *E. coli* as 130-kDa protoxin inclusions at levels comparable to the Cry4Aa wild-type (Wt) toxin (Figure 8A, Inset). Experiments were subsequently performed to assess the in vitro solubility of mutant protoxin inclusions in the carbonate buffer (pH 9.0) in comparison with that of the Wt inclusion. As the amounts of the 130-kDa soluble proteins in the supernatant were compared with those of the proteins initially used, the results revealed that all the three mutant toxin inclusions exhibited good solubility comparable to the Wt inclusion, giving >90% solubility, suggesting their properly folded conformational quality. 

When *E. coli* cells expressing each Cry4Aa-loop mutant were assayed for their biotoxicity against *C. quinquefasciatus* larvae, only the K514A mutant showed a large decrease in larvicidal activity (only ~40% mortality) while the two remaining mutants (T512A and Y513A) still retained their high larval toxicity (>90% mortality) at levels comparable to Wt (Figure 8A).It is worth mentioning that Cry4Aa-Lys^514^ is just positioned relative to Cry4Ba-Asp^454^ placed within the β10-β11 loop (see Figure 8B). Further substitution of Lys^514^ with Asp (K514D) was thus performed and revealed that its larvicidal activity was further reduced to <10%, indicating a much adverse influence of a negatively charged residue at this critical binding position. Furthermore, in silico attempts have been made in order to calculate the binding affinity change (ΔΔG_bind_) in Cry4Aa-*Cq*-mALP1264 interactions upon Ala-substitutions. Such calculation revealed that the K514D mutation displayed the largest change of binding free energy (0.85 kcal/mol) as compared to three other mutations, T512A, Y513A and K514A, giving ΔΔG_bind_ of 0.10, 0.39 and 0.56 kcal/mol, respectively (see Figure 8C). 

## 3. Discussion

Thus far, besides cadherin-like proteins and GPI-linked APNs, several mALPs have been identified as functional Cry toxin receptors in various species of insect larvae, including in mosquito larvae such as Cry4Ba and Cry11Aa receptors from *A. aegypti* [36,37,50,51] and Cry11Ba receptors from *A. aegypti* and *An. gambiae* [52,53]. In the present study, two full-length ALP homoloques from *C. quinquefasciatus*, designated as *Cq*-mALP1263and *Cq*-mALP1264, were cloned and successfully expressed in *Sf*9 non-target insect cells as 57- and 61-kDa membrane-bound proteins, respectively. Multi-approach bioinformatics analysis disclosed that both *Cq-*mALP homoloques share significant sequence similarity to *A. aegypti*-mALP—a Cry4Ba toxin receptor, displaying several conserved regions of mALP features.

It is worth noting that the detected sizes of the expressed protein bands actually corresponded to both predicted protein sizes (57-kDa *Cq*-mALP1263 and 61-kDa *Cq*-mALP1264), suggesting that both recombinant *Cq*-mALP proteins were unlikely to be glycosylated, at least, in the *Sf9* cells, albeit both ALP isoforms were predicted to contain a potential N-glycosylated site and several O-glycosylated sites. Previously, a similar observation was found for another Cry4Ba receptor, *i.e*. *Aa*-mALP, which was also found to be non-glycosylated in *Sf9* cells [54] and could retain high-binding affinity for the Cry4Ba toxin when heterologously expressed in *E. coli* [44]; this notion also agrees well with other studies, which suggested that an *E.coli* expressed-*An. gambiae-*ALP isoform (*Ag*-ALP1t) was least likely to be glycosylated and thus, that its counterpart toxin—Cry11Ba, was believed to recognize the polypeptide sequence of *Ag*-ALP1t rather than a saccharide moiety [52].

Upon qualitative and quantitative cytotoxicity evaluation, differential susceptibility to both toxins was clearly revealed for only *Cq-*mALP1264-expressedcells that displayed greater susceptibility to Cry4Aa than Cry4Ba. However, immunolocalization assays indicated that the two toxins were capable of binding to both *Cq-*mALP isoforms, which were predominantly expressed and localized on the cell membrane surface, albeit a much stronger binding intensity was found for Cry4Aa; these results clearly indicated that both mosquito-active toxins could exploit only the 61-kDa *Cq*-mALP1264 isoform, one of the two cloned *Cq*-mALP isoforms, as a functional receptor in mediating cell cytolysis. In addition, the cytotoxicity results also suggested that the binding of each individual toxin to their single functional counterpart—*Cq*-mALP1264 is sufficient for mediating the toxin activity without the requirement of pre-interaction with another receptor. Our results are rather different from other studies reported previously for lepidopteran-specific Cry1A toxins that pre-binding of the toxin to cadherin-like receptors seems to be critically needed for toxin oligomerization before interacting with second membrane-bound receptors (e.g., mALPs or mAPNs) for membrane insertion and pore formation [7,55]. However, it has been recently claimed that the completion of the oligomeric assembly of Cry1A-ABCC2 receptor complexes could directly signify membrane insertion and pore formation without the pre-binding to cadherin receptors [56]; it is noteworthy from the cytotoxicity studies that the *Cq*-mALP1264-expressed *Sf*9 cells were found to be much more susceptible to Cry4Aa than Cry4Ba; these in vitro cytotoxic effects are thus in good agreement with our previous biotoxicity findings that Cry4Aa is much more active than Cry4Ba against *Culex* larvae [4]. 

Since there is no experimentally resolved 3D structure of the *Cq*-mALP receptor or other insect ALPs, a plausible homology-based model of both identified *Cq*-mALP isoforms was herein built based on the highest sequence similarity of the known shrimp-ALP structure [57]. The Phi/Psi values observed in both modeled structures signify that such obtained *Cq*-mALP models would remain in sterically favorable main-chain conformations. Through successive docking of each individual mosquito-active toxins with their functional counterpart isoform—*Cq*-mALP1624, we have created a preferred orientation of both docking complex structures, which could infer potential binding residues confined in different domains, DII for Cry4Aa whereas DIII for Cry4Ba, to be implicated in toxin-receptor interactions. Such docking results may perhaps reflect the differential cytotoxic action of the two toxins on the *Cq*-mALP1264-expressed *Sf*9 cells. Likewise, these mosquito-active toxins could conceivably exploit different domains in functional binding to a particular membrane-bound ALP isoform—*Cq-*mALP1264 for mediating their differential toxicity against *Culex* larvae; it should be noted that when each of the two toxins was docked to *Cq*-mALP1263, both toxins still made use of different domains (i.e., Cry4Aa-DIIand Cry4Ba-DIII) in the binding to such a non-functional ALP isoform as similar to their binding to the *Cq-*mALP1264 isoform. Additionally, the number of potential interacting residues in both toxin-*Cq*-mALP1263 complexes is significantly fewer than that of the toxin-*Cq*-mALP1264 complexes, being consistent with the immunolocalization results, which revealed an obviously lower fluorescence intensity for both toxins in binding to *Cq*-mALP1263 as compared to their interactions with the functional isoform.

Subsequent mutagenic analysis of three particular surface-exposed loop residues, i.e., Thr^512^, Tyr^513^ and Lys^514^, located on the β10-β11loop in the receptor-binding domain—DII, clearly revealed that only the K514A mutant exhibited a severe reduction in toxicity against *C. quinquefasciatus* larvae. Thus, these results suggested that DII-Lys^514^is basically involved in Cry4Aa activity against the target mosquito larvae, conceivably being exploited for the toxin binding to *Cq*-mALP1624; it is nevertheless noteworthy to mention that not all the selected receptor-interacting residues predicted by such in silico docking had displayed an impaired effect on toxin larval toxicity when they were individually mutated to Ala. However, our present findings are rather contrary to other related studies where Ala-substitutions in the Cry4Aa β10-β11loop at Thr^512^ and Tyr^513^, but not Lys^514^, were claimed to have an negative effect on toxin activity against *C. pipiens* mosquito-larvae [20]. The discrepancy of such mutational effects observed between *C. quinquefasciatus*and *C. pipiens* might reflect different types of target receptors for the Cry4Aa toxin present in these two *Culex* species; it is also worth mentioning that the positions of Cry4Aa-Lys^514^ and Cry4Ba-Asp^454^are exactly stacked on each other (see Figure 8B).Additional replacement of this critical Cry4Aa-loop residue—Lys^514^ with Asp (K514D) revealed a further reduction in the larvicidal activity of the Cry4Aa toxin, indicating an adverse influence of a negative charge at this critical receptor-binding position. Nevertheless, our previous replacement of this critical residue with Asn (K514N) showed no effect on toxicity against *C. quinquefasciatus* larvae [16], suggesting that the uncharged polar feature of an Asn side-chain could compensate for the positively charged characteristic of the Cry4Aa-Lys^514^ residue.

As at this stage, we still could not obtain a purified functional *Cq*-mALP1264 isoform to be used for binding studies, in silico attempts were therefore made instead to calculate ΔΔG_bind_ for determining affinity binding changes in Cry4Aa-*Cq*-mALP1264 interactions upon such single substitutions. The computed results revealed that the K514D mutation showed the largest ΔΔG_bind_ value or the greatest decrease in binding affinity as compared to three other Ala-substitutions (i.e., T512A, Y513A, K514A), signifying an impact of a negative charge at this position on *Cq*-mALP1264-binding; this perception seems to be consistent with our previous findings for an negatively charged impact ofCry4Ba-Asp^454,^ which can be superimposed with Cry4Aa-Lys^514^, seeing that a large increase in the Cry4Batoxicity against *Culex* larvae was achieved toward an opposite-charge conversion of Asp^454^ to Lys or Arg [25]. In this context, when the Cry4Ba-D454K modeled structure was docked with *Cq*-mALP1264, the mutant toxin appeared to change its binding orientation, being able to utilize DII instead of DIII particularly through the mutated residue—Lys^454^ along with other residues in β2-β3 (Tyr^332^ and Asp^334^) and β10-β11 (Tyr^455^) loops (see Appendix A).

Taken together, our present study has disclosed that these two mosquito-active toxins, Cry4Aa and Cry4Ba, conceivably exploited different domains in functional binding to the same *Culex* membrane-bound ALP isoform—*Cq-*mALP1264 for mediating differential toxicity against *Culex* target larvae. Our data also suggested that a positively charged side-chain near the tip of the β10-β11 loop within DII would play a critical role in determining a degree of difference in larvicidal activity of these two closely-related toxins against *Culex* larvae.

## 4. Materials and Methods 

### 4.1. Cloning of GPI-Anchored ALP-Coding Regions from C. quinquefasciatus Larval Midgut Transcripts

Total midgut RNAs were extracted from 4^th^ instar *C. quinquefasciatus* larvae using TRI Reagent^®^ (Molecular Research Center, Inc., Cincinnati, OH, USA) and subsequently reverse-transcribed into first-strand complementary DNAs (cDNAs) by using Improm-II^™^ reverse transcriptase (Promega, Madison, WI, USA) together with oligo-d(T) primers. The first strand cDNAs were then used for amplifying the two putative full-length *Cq*-mALP cDNAs using each set of *Cq*-mALP specific primers (i.e., *Cq*-mALP1263-*f*, 5′-CACCATGCAGTTGCTTGCAG TAGTTAC-3′ and *Cq*-mALP1263-*r*,5′-TCATCACGTGCATGCCTTCAAG-3′; *Cq*-mALP1264-*f*,5′-CA CCATGAAGACCATCTTGCTGGT A-3′ and *Cq*-mALP1264-*r*,5′-CTACTACACCTTGATCCAGCTC G-3′) and Phusion™ DNA polymerase (Thermo Fisher Scientific, Waltham, MA, USA). The amplicons with the expected size were purified and ligated into pENTR^TM^/SD-TOPO^®^ (Invitrogen, Carlsbad, CA, USA) to produce recombinant transfer pENTR/*Cq*-mALP plasmids. The recombinant clones were isolated from transformed One Shot^™^ TOP10 competent cells and subjected to DNA sequencing analysis. 

### 4.2. Bioinformatics and Sequence Analysis of the Cloned Cq-mALPs

Sequence homology of the two full-length deduced amino acid sequences ofCq-mALPs was determined using BLAST tool online services (http://www.ncbi.nlm.nih.gov/BLAST (accessed on 12 September 2022)). Sequence alignments of *Cq-*mALPs with other mALPs, i.e., *Px*-mALP (AHF20243.2), *Aa*-mALP (ACV04847.1), *Bm*-mALP (NP_001037536.3), *Sf*-mALP (ALS30430.1), *Ag*ALP1 (AGN95448.1), *Hv*-mALP (ACP39712.1), *Rn*-rIAP (NP_073156.3) and *Hm*-PLAP (NP_112603.2), were performed by ClustalX and displayed with GeneDoc. Nucleotide sequences of *cq-mALP1263* and *cq-mALP1264* have been deposited in GenBank database under the accession numbers OM864012 and OM864013, respectively. Prediction of signal peptide sequence of both *Cq*-mALP deduced amino acid sequences was performed online using three programs, including SignalP-5.0 (https://services.healthtech.dtu.dk/service.php?SignalP-5.0 (accessed on 12 September 2022)), NetNGlyc-1.0 (https://services.health tech.dtu.dk/service.php?NetNGlyc-1.0 (accessed on 12 September 2022)) and NetOGlyc-4.0 (https://services.healthtech.dtu.dk/ service.php?NetOGlyc-4.0 (accessed on 12 September 2022)). Their GPI-anchoring sites were predicted and analyzed by four different programs, including PredGPI (http://gpcr.biocomp.unibo.it/predgpi/ (accessed on 12 September 2022)), big-PI Predictor (https://mendel.imp.ac.at/gpi/gpiserver.html (accessed on 12 September 2022)), NetGPI-1.1 (https://services.healthtech.dtu.dk/service.php?NetGPI (accessed on 12 September 2022)) and GPI-SOM (http://gpi.unibe.ch/ (accessed on 12 September 2022)).

### 4.3. Expression of Recombinant Cq-mALP Proteins in Sf9 Insect Cells

*Spodoptera frugiperda* (*Sf*9) cells were maintained in Sf-900^™^ culture medium (Gibco, Fisher Scientific, Pittsburgh, PA, USA) supplemented with 10% fetal bovine serum (Life Technologies, Carlsbad, CA, USA) and 5% penicillin/streptomycin. Cellswere grown as a monolayerin 25-cm^2^ culture flasksat26 °C until their reached 80–90% confluence. The number of cells was determined by hemocytometer (Hausser Scientific, Horsham, PA USA) with trypan blue exclusion assay.

Each recombinant pENTR/*Cq*-mALP plasmid was selected for an in vitro Lamda Recombination. Individual *Cq*-mALP fragments were transferred from pENTR/*Cq*-mALP plasmids to Baculodirect Linear DNA (AcMNPV) by site-specific recombination. Both resulting constructs, AcMNPV/ *Cq*-mALP1263 and AcMNPV/*Cq*-mALP1264, containing each corresponding *Cq*-mALP sequence under the control of the polyhedrin promoter were transfected into insect *Sf*9 cells by cationic liposome-mediated gene transfer method. In brief, *Sf*9 cells were counted and seeded at a density of 8×10^5^ cells/well into 6-well culture plates. Cells were co-transfected with a mixture containing each recombinant transfer pENTR/*Cq*-mALP plasmid, linearized baculovirus C-term DNA and Cellfectin™ reagent in Grace’s insect medium (Gibco, Fisher Scientific, Pittsburgh, PA, USA). Following 5-h incubation, the transfection mixture was replaced with a complete growth medium containing 5% (*v*/*v*) fetal bovine serum and transfected cells were allowed to grow for 5 days at 26 °C. 

For production of recombinant proteins, the monolayer cultures were infected with viral solution for 3 days at 26 °C. Then cells were washed and harvested by centrifugation. The cell pellet was resuspended in PBS (phosphate-buffered saline, pH 7.4) and analyzed for expression of recombinant *Cq*-mALP proteins by SDS-PAGE. In the experiments, *Sf*9 cells expressing individual recombinant plasmids encoding either CAT or *Aa*-mALP were used as negative and positive controls, respectively. Furthermore, the protein bands corresponding to the *Cq*-mALP proteins were excised and subsequently verified for their identities by mass spectrometry (Proteomics International Laboratories LTD, Nedlands, WA, Australia).

### 4.4. Determination of ALP Activity in Cq-mALP-Expressed Cells

The standard assay of ALP activity was carried out following the method as previously described [54]. After 3 days post-infection, *Sf*9 cells expressing recombinant *Cq*-mALP proteins were collected by centrifugation and washed twice with PBS, pH 7.4. Cell lysates were resuspended in ALP buffer (100 mΜ Tris-HCl, pH 9.5, 5 mΜ MgCl_2_ and 100 mΜ NaCl) containing 5 mΜpNPP (New England BioLabs, Ipswich, MA, USA) as an ALP substrate. The reactions were incubated at 30 °C for 10 min and terminated by the addition of 1 N NaOH. One unit of ALP enzyme can hydrolyze 1 μmole of pNPP in 1 min at 30 °C.The hydrolysis of colorless pNPP to chromogenic p–nitrophenol (pNP) product was determined by spectrophotometer at 405 nm. The analysis of hydrolytic reactions was performed in triplicate.

### 4.5. Expression and Purification of Activated Cry4 Toxins

Two mosquito-active single-mutant toxins, Cry4Aa-R235Q and Cry4Ba-R203Q, were expressed and purified as described previously with slight modifications [16,58].Briefly, *Escherichia coli* strain JM109 harboring each of the Cry4-related mutant protein was grown in Luria-Bertani medium containing 100 μg/mL ampicillin at 37 °C until OD_600_ reached 0.3–0.6. The expression of Cry4 proteins was induced with 0.1 mM isopropyl-β-D-1-thiogalactopyranoside (IPTG) for 4 h and harvested by centrifugation at 6000× *g* for 10 min at 4 °C. The cell pellet was re-suspended in 100 mM KH_2_PO_4_ (pH 5.0) containing 0.1% Triton X-100 and 0.5% NaCl, and was then disrupted by sonification (Sonics Vibra-Cell^TM^ VCX750 Ultrasonic Processor, Sonics & Materials, Newtown, CT, USA). Protoxin inclusions were then collected, washed and subsequently solubilized in carbonate buffer (50 mM Na_2_CO_3_/NaHCO_3_, pH 9.0) for 1 h at 37 °C. The solubilized Cry4 proteins (~1 mg/mL) were activated by trypsin (tolylsulfonyl phenylalanyl chloromethyl ketone-treated trypsin, Sigma-Aldrich, Burlington, VT, USA) at an enzyme/toxin ratio of 1:10 (*w*/*w*). The 65-kDa of activated Cry4Aa and Cry4Ba toxins was subsequently purified by size-exclusion chromatography (Superose^®^ 12 10/300 GL, GE Healthcare, Chalfont Saint Giles, UK) being eluted with the carbonate buffer (pH 9.0) at a flow rate of 0.4 mL/min. The purified Cry4 proteins were then concentrated with a 10-kDa MWCO Millipore membrane (Merck KGaA, Frankfurter, Darmstadt, Germany). The concentration of Cry4 proteins was quantified by the Bradford assay and resolved on SDS-PAGE.

### 4.6. Cytotoxicity Assays of Cry4 Toxins on Sf9 Cells Expressing Cq-mALPs

5-day post-infected *Sf*9 cells expressing individual recombinant *Cq*-mALPs were seeded at a density of 8 × 10^5^ cells/well into 6-well plates and then treated with 50 μg/mL of either activated Cry4Aa or Cry4Ba toxin for 1 h at 25 °C. The morphology of Cry4-treated cells was then visualized and photographed under an inverted light microscope (Nikon Eclipse TS100, Nikon Instruments, Melville, NY, USA). The treated cells were harvested, washed with PBS, and subsequently evaluated for viability by measuring their metabolic activity through the reducing potential of viable cells using resazurin-based PrestoBlue^TM^(Invitrogen, Carlsbad, CA, USA) assay according to the manufacturer’s instructions. Since the conversion of resazurin into resorufin requires the reducing environment of the viable cells, relative cell viability was calculated by measuring the resorufin absorbance at 590 nm (excitation at 530 nm). Statistical analysis was performed by one-way ANOVA followed by the Bonferroni Post-Hoc test for multiple comparisons by using the SPSS 27.0 software(IBM Corp., SPSS Statistics Software, Armonk, NY, USA). Statistical significance was considered at ∗ *p* < 0.05.

### 4.7. Immuno-Fluorescence of Cry4 Toxins Binding to Cq-mALP-Expressing Cells

After 5-day post-infection, *Sf9* cells expressing *Cq*-mALPs (*Cq*-mALP1263 or *Cq*-mALP1264), CAT or *Aa*-mALP were incubated with 50 μg/mL of either activated Cry4 toxin at 26 °C for 1 h. Cry4-treated cells were washed twice with PBS and collected by centrifugation at 2000× *g* for 5 min. The cell pellets were fixed with ice-cold 4% paraformaldehyde for 30 min at 4 °C and washed twice with PBS, prior to blocking in 5% bovine serum albumin (BSA) for 1 h at 4 °C. Fixed cells were incubated with primary antibodies specific to each Cry4 toxin for 1 h at 4 °C. After washing with PBS thoroughly, the cells were incubated with rhodamine-conjugated (for Cry4Aa) or fluorescein-conjugated (for Cry4Ba) secondary antibodies (Amersham Life Science, Arlington Heights, IL, USA) for 1 h at 4 °C. The stained cells were rinsed with PBS and mounted on microscope slides, followed by visualized with the confocal laser scanning microscope (Olympus FV1000 LS, Olympus, Shinjuku, Tokyo, Japan), using Fluoview (FV10-ASW) Viewer 4.1 software. 

### 4.8. Homology-Based Modeling and Validation of Cq-mALP Structures

Plausible 3D modeled structures of *Cq*-mALPs were constructed based on the target-template alignment with the shrimp-ALP high-resolution (1.9Å) crystal structure (PDB: 1K7H) [56] *via* Phyre2 (http://www.sbg.bio.ic.ac.uk/phyre2/ (accessed on 12 September 2022)) and displayed by UCSF Chimera (https://www.cgl.ucsf.edu/ chimera/ (accessed on 12 September 2022)). The local quality of both modeled structures was validated by VERIFY-3D [59] and their stereochemistry was analyzed using the Psi/Phi Ramachandran plot computed with PROCHECK [60]. The root mean square deviation (RMSD) of each *Cq*-mALP model to the shrimp-ALP template structure was calculated by structural superposition using Chimera Tool [61].

### 4.9. In Silico Intermolecular Docking between Cq-mALP1624 and Individual Cry4 Toxins

Molecular docking was done using ClusPro 2.0 [62] in order to identify the binding sites between the functional toxin receptor—*Cq*-mALP1264 model and each individual Cry4 toxin structure, Cry4Aa-R235Q (PDB: 2C9K) or Cry4Ba-R203Q(PDB: 4MOA) as their coordinate files were generated by SWISS-MODEL (http://swissmodel.expasy.org (accessed on 12 September 2022)). Docking calculations were performed using ClusPro 2.0 through a global soft rigid body search program (PIPER). The best docking complex structure was selected and subjected to energy minimization. Bonding between interacting residues was analyzed as hydrogen bonding, which was calculated based on the criteria of 2.7 Å for hydrogen-accepter distance and 3.3 Å for donor-acceptor distance using LIGPLOT 1.4.4 [63]. A possible effect on binding affinity of selected mutations on toxin-receptor interactions of each mutant-receptor docking complex was analyzed by computing the binding affinity changes (ΔΔG_bind_) using the MutaBind2 program [49].

### 4.10. Construction, Expression and Solubilization of Cry4Aa Mutant Toxins

The p4Aa-R235Q plasmid, encoding the 130-kDa Cry4Aa-R235Q protoxin [16], was used as a template together with individual pairs of complementary mutagenic primers designed for single substitutions of selected β10–β11 loop residues. All mutant plasmids were generated by polymerase chain reaction using high-fidelity Phusion DNA polymerase (Finnzymes, Espoo, Southern Finland, Finland), following the QuickChange^TM^ mutagenesis procedure (Stratagene, Santa Clara, CA, USA). Mutant plasmids in selected *E. coli* clones were initially identified by restriction endonuclease analysis and subsequently verified by DNA sequencing. All mutant toxins were over-expressed in *E. coli* strain JM109 as inclusions upon 4-h induction with isopropyl-β-D thiogalactopyranoside (IPTG, 0.1 mM final concentration). Individual toxin inclusions (~1 mg/mL) were solubilized in 50 mM Na_2_CO_3_/NaHCO_3_ (pH 9.0) at 37 °C for 1 h as previously described [16].Following centrifugation (6000× *g*, 10 min, 4 °C), supernatants containing soluble toxins were subjected to SDS-PAGE analysis in comparison with their corresponding inclusion suspensions.

### 4.11. Biotoxicity Assays of Cry4Aa-R235Q and Its Mutant Toxins 

Bioassays for larvicidal activity of *E. coli* cells expressing Cry4Aa-R235Q (Wt) or its mutant toxins were performed against *C. quinquefasciatus* mosquito larvae as described elsewhere [13]. Cells containing pUC12 vector plasmid were used as a negative control. Mortality was recorded after 24-h incubation period and three independent experiments were performed for each toxin. Statistical analysis was carried out by using one-way ANOVA followed by Bonferroni Post-Hoc test for multiple comparisons. 

## Figures and Tables

**Figure 1 toxins-14-00652-f001:**
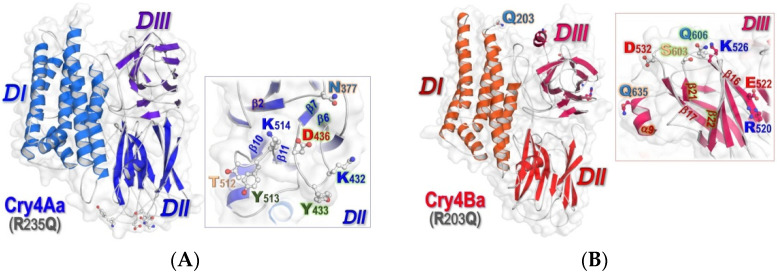
Combined surface-ribbon representations of the 65-kDa crystal structures of two closely related Cry4 mosquito-active toxins, illustrating their three-domain organization (DI-DIII). (**A**) Cry4Aa-R235Q (PDB: 2C9K [13]) and (**B**) Cry4Ba-R203Q (PDB: 4MOA [14]), Insets, zoomed in views of potential receptor-binding residues, which are situated in each different domain (Cry4Aa-DII and Cry4Ba-DIII) are represented as ball-and-stick models.

**Figure 2 toxins-14-00652-f002:**
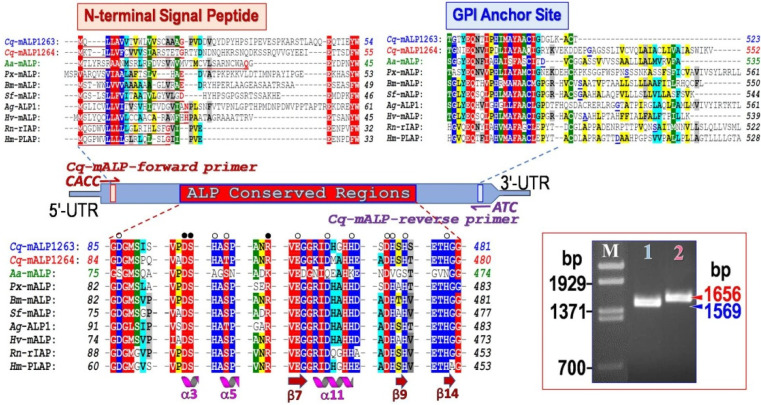
Multiple alignments of deduced amino acid sequences of *Cq*-mALP1263and *Cq*-mALP1264 against other identified mALP sequences, including *Rn*-rIAP (*Rattus norvegicus*), *Hm*-PLAP (*Homo sapiens*), *Aa-*mALP (*A. aegypti*), *Px*-mALP (*Plutellaxylostell*a), *Bm*-mALP (*Bombyx mori*), *Sf*-mALP (*S. frugiperda*), *Ag-*ALP1 (*Anopheles gambie*) and *Hv*-mALP (*Heliothisvirescens*). Amino acids are shaded red, blue, green, cyan and yellow to denote degree of homology (10/10), (9/10), (8/10), (7/10) and (6/10), respectively. Signal peptide sequences are underlined along with their predicted cleavage sites denoted by red underlined letters. GPI anchor sites are indicated by blue underlined letters. Residues participating in the ALP active site, and metal- or substrate-binding sites are denoted by opened and filled circles, respectively. Secondary structure elements are shown under the *Hm*-PLAP reference sequence. Inset, RT-PCR analysis of total RNAs extracted from *C. qiunquefasciatus* larval midgut, showing the cDNAs of the two full-length transcripts, *Cq*-mALP1263 (*lane 1*) and *Cq*-mALP1264 (*lane 2*), as their sizes indicated.

**Figure 3 toxins-14-00652-f003:**
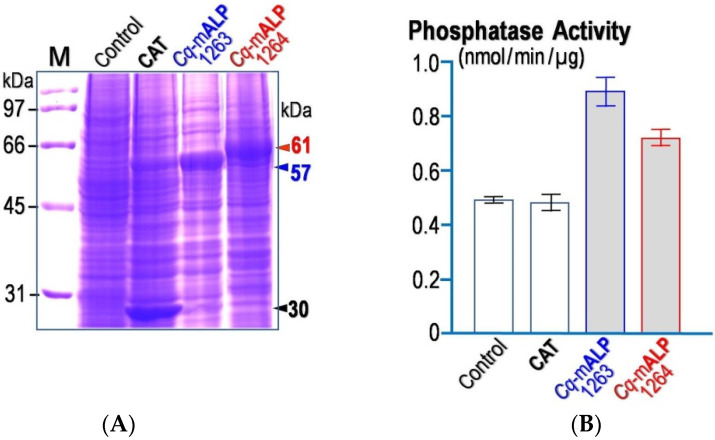
(**A**)SDS-PAGE analysis (Coomassie brilliant blue-stained 10% gel) of *Sf*9 cell lysates expressing *Cq*-mALP1263or *Cq*-mALP1264 in comparison with those expressing CAT as indicated. Control represents the non-infected cell lysate. M, broad-range protein markers. (**B**) ALP activity through pNPP hydrolysis of *Sf*9 cell lysates expressing each individual *Cq*-mALP was assessed in comparison with that of CAT-expressed cell lysate, and of the non-infected cell lysate. Error bars indicate standard errors of the mean (SEM) from three independent experiments.

**Figure 4 toxins-14-00652-f004:**
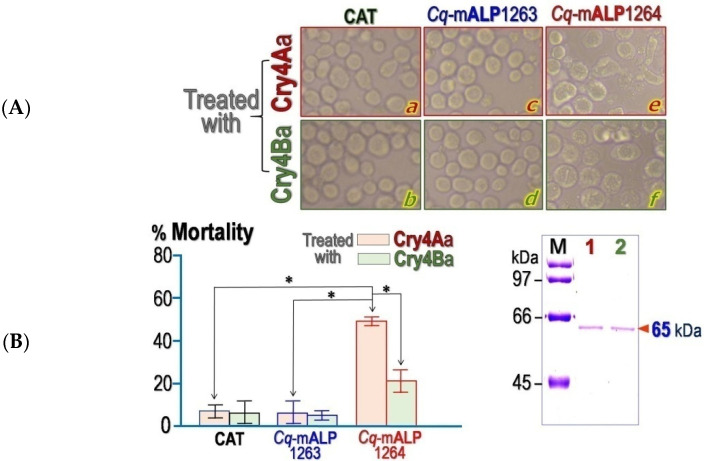
Cytotoxic effects of Cry4 toxins on *Cq*-mALPs-expressed cells. (**A**) Morphological features of *Cq*-mALP-expressed cells in the presence of Cry4 toxins. Representative images are shown for Cry4Aa-treated cells (*upper panels*) and Cry4Ba-treated cells (*lower panels*). Morphological changes of Cry4-treated cells were visualized and photographed under an inverted microscope for CAT-expressed cells (**a**,**b**), *Cq*-mALP1263-expressed cells (**c**,**d**) and *Cq*-mALP1264-expressed cells (**e**,**f**). All images shown are representatives of those seen from at least two such fields of view per sample and three independent experiments. (**B**) Percentage of cell mortality, as assayed using PrestoBlue^™^, of *Sf*9 cells expressing each of *Cq*-mALPs or CAT. Each bar represents SEM determined from triplicate samples of three independent experiments. Statistical significance was considered at ∗ *p*< 0.05. Inset, SDS-PAGE analysis (Coomassie brilliant blue-stained 10% gel) of the activated Cry4 toxins. Lanes 1 and 2 are trypsin-activated products of Cry4Aa-R235Q and Cry4Ba-R203Q toxins, respectively. The arrow indicates the band corresponding to the 65-kDa purified toxins. Lane M represents the broad-range protein markers.

**Figure 5 toxins-14-00652-f005:**
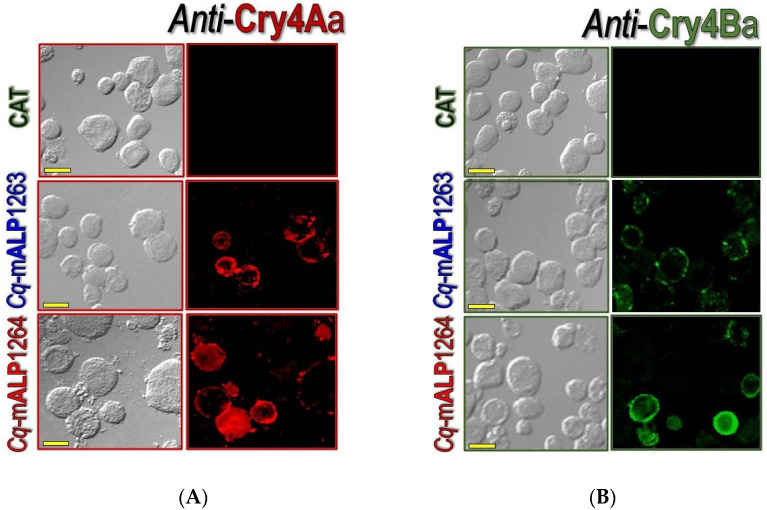
Immuno-staining of *Cq*-mALP-expressed cells against Cry4 toxins. Cells were treated with 50 µg/mL activated toxins of (**A**) Cry4Aa or (**B**) Cry4Ba for 1 h. Formaldehyde-fixed cells were labeled with each primary antibody directed against Cry4Aa or Cry4Ba, and further stained with rhodamine-conjugated secondary antibody (for Cry4Aa) and fluorescein-conjugated secondary antibody (for Cry4Ba). All images shown are representatives of those seen from several fields of view per sample and at least two independent experiments. Scale bars represent 20 μm.

**Figure 6 toxins-14-00652-f006:**
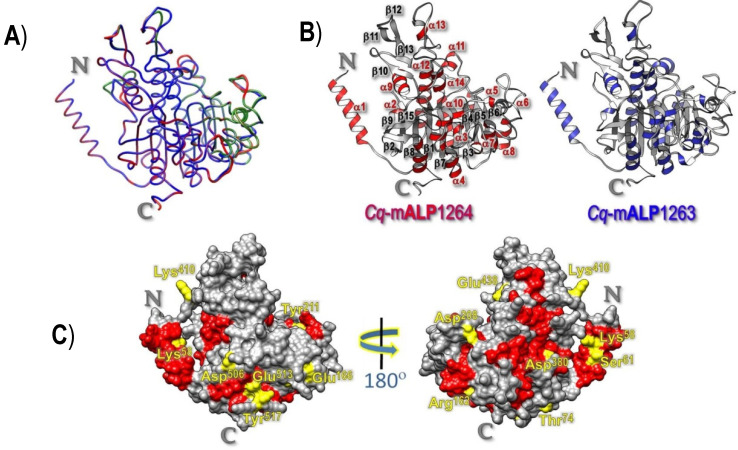
Homology-based 3D modeled structure of *Cq*-mALPs. (**A**) Superposition of Cα traces of *Cq*-mALP1263 (*blue*) and *Cq*-mALP1264 (*red*) models with the shrimp ALP structure (*green*) prepared *via* Chimera 1.7. (**B**) Ribbon representation of *Cq*-mALP1263 (*right panel*) and *Cq*-mALP1264 (*left panel*) modeled structures, illustrating ten β-strands (*gray*) surrounded by α-helices (*red or blue*) in the core structure. (**C**) Surface representation of the *Cq*-mALP1264 modeled structure, illustrating several uncharged-polar and charged surface-exposed residues preserved exclusively in *Cq*-mALP1264 (*yellow*) along with other common surface-exposed residues located in α-helices (*red*) and β-strands with connecting loops (*gray*).

**Figure 7 toxins-14-00652-f007:**
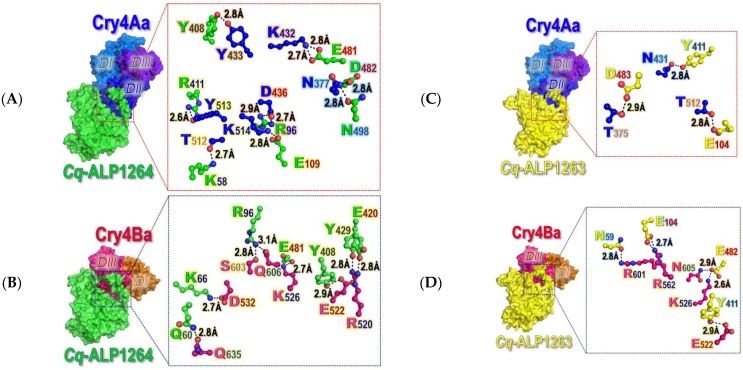
Surface representations of the resultant docking complexes, illustrating the three-domain Cry4 mosquito-active toxins, either Cry4Aa-R235Q or Cry4Ba-R203Q, interacting with their functional receptor—*Cq*-mALP1264 (**A**,**B**) or non-functional receptor—*Cq*-mALP1263 (**C**,**D**). Insets, zoomed in views of potential receptor-binding residues in Cry4Aa-DII and Cry4Ba-DIII, which are represented as ball-and-stick models along with their interacting partner residues.

**Figure 8 toxins-14-00652-f008:**
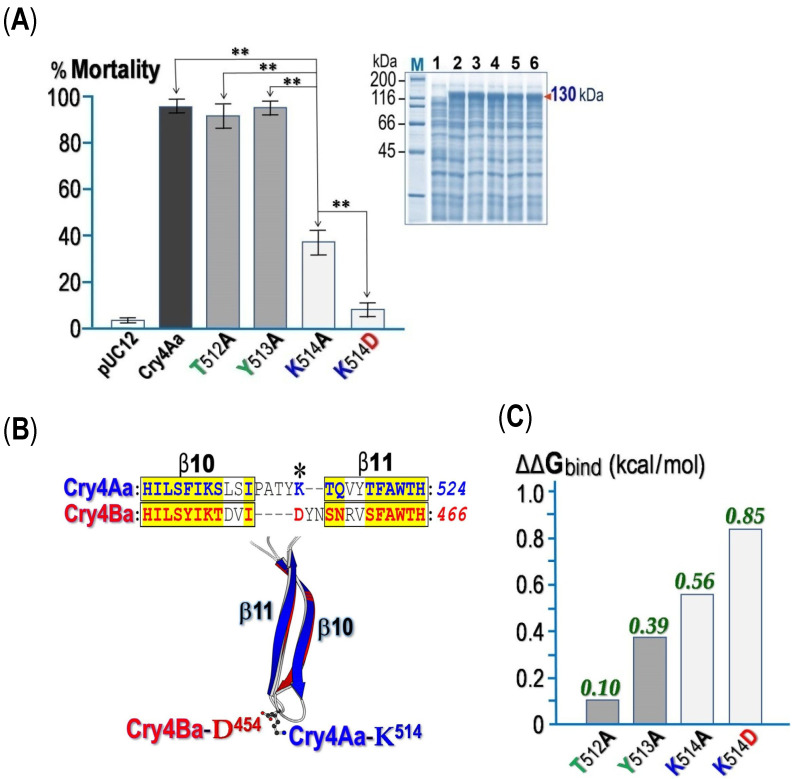
(**A**) Larvicidal activity of *E. coli* cells (~10^8^ cells/mL) expressing the Cry4Aa or its loop mutants (T512A, Y513A, K514A and K514D) tested against *C. quinquefasciatus* larvae. Cells containing the pUC12 plasmid vector were used as a negative control. Error bars indicate SEM from at least three independent experiments. Statistical significance was considered at ∗∗ *p*< 0.01. Inset, SDS-PAGE (Coomassie brilliant blue-stained 10% gel) analysis of the lysate extracts of *E. coli* cells (~10^7^ cells)expressing the130-kDa Cry4Aa (*lane 2*) or its mutant protoxins (*lanes 3–6* denote T512A, Y513A, K514A and K514D, respectively). Lane 1 represents the lysate extract containing pUC12. Lane M represents the broad-range protein markers. (**B**) *Upper panel*, pairwise sequence alignment derived from known structures of the β10-β11 receptor-binding loop of Cry4Aa and Cry4Ba. Corresponding β-strands of both toxins are denoted with rectangles. *Lower panel*, superimposition of the β10-β11 hairpin from Cry4Aa (*shaded blue*) and Cry4Ba (*shaded red*) crystal structures, revealing that the positions of Cry4Aa-Lys^514^ and Cry4Ba-Asp^454^are exactly stacked on each other as indicated by * in the sequence alignment. (**C**) In silico calculation of the change of binding free energy (ΔΔG_bind_) in Cry4Aa-*Cq*-mALP1264 interactions upon each mutation as indicated using the MutaBind2 program [49].

## Data Availability

Not applicable.

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
