# Peer review of "Cry4Aa and Cry4Ba Mosquito-Active Toxins Utilize Different Domains in Binding to a Particular Culex ALP Isoform: A Functional Toxin Receptor Implicating Differential Actions on Target Larvae"

_toxins, 2022, doi:10.3390/toxins14100652_

Round 1
Reviewer 1 Report
Dear Authors the main reason and most significant this the statistical approach. For this quality of paper is not good. I propose to do statistical analysis using Kaplan-Meier and Anova with the mortality data. This statistical approach may help more the reader of the paper
Author Response
Reviewer 1:
1) The main reason and most significant is the statistical approach. For this quality of paper is not good. I propose to do statistical analysis using Kaplan-Meier and Anova with the mortality data. This statistical approach may help more the reader of the paper.
Answer: As kindly suggested, we have now performed statistical analysis of both cytotoxicity and larvicidal activity data via one-way analysis of variance (ANOVA) followed by Bonferroni Post-Hoc test for multiple comparisons by using the SPSS 27.0 software (IBM Corp., SPSS Statistics Software, Armonk, NY, USA). Statistical significance was considered at  p < 0.05.
The results have been added in the revised MS [see Fig. 4B & Fig. 8A, and the highlighted texts in lines 620-623].
On behalf of all others, I would like to express my sincere thanks to you and the reviewers for all constructive comments and suggestions. I am glad to say that we have now complied with most of the requests that the reviewers have asked explicitly.
We hope that you will find our revised paper sufficiently modified to be acceptable for publication in Toxins.
Reviewer 2 Report
This manuscript describes a series of experiments designed to investigate the differential toxicity of Cry4Aa and Cry4Ba against Culex quinquefasciatus. The results are largely informative and provide useful data on this topic. The ectopic expression studies are particularly strong and provide convincing evidence for the role of the 1264 isoform as a functional receptor. The molecular docking study is less convincing and I would like to see more controls here as outlined below.
Major issues.
1) The docking studies indicate a different binding orientation of the two toxins which the authors suggest could explain there differential toxicity. I would like to see more docking experiments done including the docking of the two toxins to the 1263 isoform - are any differences seen here that could explain why this is not a functional receptor. Since the exchange of loops (Ref20) increases the activity of Cry4Ba the structure of this mutant could be modelled and also docked to see if it changes the binding orientation.
Minor issues
2) Section 2.4 The data shown in Fig 5 are not sufficient to indicate that binding of the toxins to 1263 is less that that to 1264. We are only shown about 6 cells, either we need to be shown more cells or the authors need to quantify the binding over many more cells.
3) In several places the binding of the toxin to the cell is described as "specific". In the field of Bt toxin binding "specific" has a particular meaning - binding which is disrupted by an excess of non-labelled toxin. Since such competition studies were not performed this term should not be used. There could be non-specific binding to the transgenic ALP.
4) Line 253 delete "as expected". You should never expect a result.
5) In the discussion (lines 444-451) the authors make reference to the sequential binding model. This model is going out of favour and the authors should acknowledge this in the context of a recent publication doi.org/10.3390/toxins14070433
Author Response
Reviewer 2:
Major issues
1) The docking studies indicate a different binding orientation of the two toxins which the authors suggest could explain their differential toxicity. I would like to see more docking experiments done including the docking of the two toxins to the 1263 isoform- are any differences seen here that could explain why this is not a functional receptor. Since the exchange of loops (Ref20) increases the activity of Cry4Ba the structure of this mutant could be modeled and also docked to see if it changes the binding orientation.
Answer: As kindly suggested, we have now performed more docking analysis of the two toxins with the CqmALP1263 isoform. It was found that both toxins still exploited of different domains (i.e. Cry4Aa-DII and Cry4Ba-DIII) in the binding to such a non-functional ALP isoform. The results have been added in the revised MS [see Fig. 7C & D, and the highlighted texts in lines 343-347 and 479-486].
In addition, we have done the toxin-receptor docking of our own single-mutant toxin (i.e. Cry4Ba-D454K) instead of one of the exchanged-loop mutants (i.e. 4BL3-PAT in Ref. 20 as suggested by reviewer) since such an exchanged-loop mutant does NOT include a positively charged side-chain in the loop 3 (the β10-β11 loop) unlike ours. When the D454K modeled structure was docked with Cq-mALP1264, the mutant toxin appeared to change its binding
orientation, being able to utilize DII instead of DIII particularly through the mutated residue—Lys454 along with other residues in β2-β3 (Tyr332 and Asp334) and β10-β11 (Tyr455) loops [see Supplementary Fig. S2, and the highlighted texts in lines 515-519].
Minor issues
2) Section 2.4 The data shown in Fig. 5 are not sufficient to indicate that binding of the toxins to 1263 is less than that to 1264. We are only shown about 6 cells, either we need to be shown more cells or the authors need to quantify the binding over many more cells.
Answer: Many thanks for the reviewer’s concern. In fact, all the cell images shown are representatives of those seen from several fields of view per sample and at least two independent experiments as now mentioned in the revised MS [highlighted in lines 282-283].
3) In several places the binding of the toxin to the cell is described as "specific". In the field of Bt toxin binding "specific" has a particular meaning - binding which is disrupted by an excess of non-labeled toxin. Since such competition studies were not performed this term should not be used. There could be non-specific binding to the transgenic ALP.
Answer: For such a concern, we have now removed this word from the revised MS as highlighted in lines 31, 257, 261, 305, 450 and 455.
4) Line 253 delete "as expected". You should never expect a result.
Answer: Done as suggested.
5) In the discussion (lines 444-451) the authors make reference to the sequential binding model. This model is going out of favour and the authors should acknowledge this in the context of a recent publication doi.org/10.3390/toxins14070433
Answer: As suggested, we have now mentioned the notion of the recently published article in the revised MS as highlighted in lines 457, 460-463 and 840-841. On behalf of all others, I would like to express my sincere thanks to you and the reviewers for all constructive comments and suggestions. I am glad to say that we have now complied with most of the requests that the reviewers have asked explicitly.
We hope that you will find our revised paper sufficiently modified to be acceptable for publication in Toxins.
Round 2
Reviewer 1 Report
Accept in present form
Reviewer 2 Report
The authors have satisfactorily addressed my previous comments